# Mechanisms for a remote response to Asian anthropogenic aerosol in boreal winter

Laura Wilcox[1,2], Nick Dunstone[3], Anna Lewinschal[4], Massimo Bollasina[5], Annica M. L. Ekman[4], and Eleanor Highwood[2]

[1]National Centre for Atmospheric Science, UK
[2]Department of Meteorology, University of Reading, Reading, UK
[3]Met Office Hadley Centre, Exeter, UK
[4]Department of Meteorology, Stockholm University, Stockholm, Sweden
[5]School of Geosciences, Grant Institute, University of Edinburgh, Edinburgh, UK

**Correspondence:** Laura Wilcox l.j.wilcox@reading.ac.uk

**Abstract.** Asian emissions of anthropogenic aerosols and their precursors have increased rapidly since 1980, with half of the increase since the pre-industrial era occurring in this period. Transient experiments with the HadGEM3-GC2 coupled model were designed to isolate the impact of Asian anthropogenic aerosols on global climate in boreal winter. It is found that this increase has resulted in local circulation changes, which in turn have driven decreases in precipitation over China, alongside

5    an intensification of the offshore monsoon flow. No large temperature changes are seen over China. Over India, the opposite response is found, with decreasing temperatures and increasing precipitation. The dominant feature of the local circulation changes is an increase in low-level convergence, ascent, and precipitation over the Maritime continent, which forms part of a tropical-Pacific-wide La-Nina-like response.

HadGEM3-GC2 also simulates pronounced far-field responses. A decreased meridional temperature gradient in the North

10   Pacific leads to a positive-Pacific-North-American circulation pattern, with associated temperature anomalies over the North Pacific and North America. Anomalous northeasterly flow over northeast Europe drives advection of cold air into central and western Europe, causing cooling in this region. An anomalous anticyclonic circulation over the North Atlantic causes drying over western Europe. Using a steady-state primitive equation model, LUMA, we demonstrate that these far-field midlatitude responses arise primarily as a result of Rossby waves generated over China, rather than in the Equatorial Pacific.

## 1   Introduction

Anthropogenic aerosols account for the largest uncertainty in the radiative forcing of climate Boucher et al. (2013). Aerosols can affect climate by modulating shortwave, and to some extent, longwave radiation (aerosol radiation interactions), and through their interactions with clouds (aerosol cloud interactions). They can act as cloud condensation nuclei, which, for

constant water availability, results in a reduction in the size of cloud droplets, and an increase in cloud albedo (Twomey et al., 1984). Furthermore, smaller droplets reduce the efficiency of precipitation formation, which can lead to increases in cloud lifetime and depth (Albrecht, 1989). Absorbing aerosols introduced to a cloud layer can also cause a reduction in cloud amount by inducing local heating in the layer and cloud evaporation (semi-direct effect). Overall, increases in aerosol emissions act to
cool the climate, with a 1750-2011 radiative forcing of -0.9 (-1.9 to -0.1)W m$^{-2}$, compared to 2.83 (2.54 to 3.12)W m$^{-2}$ from greenhouse gases (Myhre et al., 2013).

    Due to numerous chemical and physical removal processes, anthropogenic aerosols only have a short residence time in the atmosphere (of the order of a few days to a couple of weeks), which causes their distribution, and the associated climate forcing, to be spatially heterogeneous. As such, aerosols can induce substantial changes in atmospheric circulation, and extend
an influence over long distances (Rotstayn and Lohmann (2002); Ramanathan et al. (2005); Allen et al. (2015)). Indeed, the influence of remote forcing can outweigh that of local forcing (Shindell et al. (2012); Lewinschal et al. (2013)). This adds additional complexity and complications in identifying the climate imprint of aerosols due to the large uncertainty in the atmospheric circulation response, especially at sub-continental scales, yet this is key to understanding drivers and projections of regional climate change.

The preferential cooling of the Northern Hemisphere, where most emissions of aerosols and their precursors are made, has been linked to a southward shift in the inter-tropical convergence zone (Hwang et al. (2013); Allen et al. (2015); Ridley et al. (2015); Allen and Ajoku (2016); Navarro et al. (2017); Voigt et al. (2017)) and a weakening of the global monsoon (Polson et al., 2014). Increases in anthropogenic aerosol have also been associated with an Equatorward shift of the Northern Hemisphere storm tracks (Kristjánsson et al. (2005); Ming and Ramaswamy (2009)), decreases in Sahel rainfall (Biasutti and
Giannini (2006); Rotstayn and Lohmann (2002)), and a weakening of the south Asian monsoon (Bollasina et al., 2011).

    Some studies have shown that the spatial patterns of temperature and precipitation responses to aerosol and precursor emission changes are similar, regardless of the emission location, as they project onto modes of climate variability (Dong et al. (2014); Dong et al. (2016); Kang et al. (2017); Kasoar et al. (2018); Westervelt et al. (2018)). Such findings are not unanimous, with other work suggesting that the spatial pattern of the aerosol-induced climate response is dependent on the emissions lo-
cation (Wang (2015); Aamaas et al. (2016)). Similarly, there is still debate about whether the climate response to increases in anthropogenic aerosols is a simple mirror of the response to increasing greenhouse gases (e.g. Feichter et al. (2004); Xie et al. (2013); Gillett et al. (2013)).

    In cases where similar patterns of climate change are found in response to aerosol from different regions, the patterns may still arise via different mechanisms depending on the emission region (Dong et al. (2014); Dong et al. (2016)). Similarly, in
cases where the circulation response to increases in aerosols and greenhouse gases appear to mirror each other, the underlying mechanisms for the change can differ. For example, Ming et al. (2011) showed that aerosols increase baroclinic instability and cause an equatorward shift in the midlatitude jet by enhancing wind shear, while greenhouse gases decrease baroclinic instability and cause a poleward shift in the jet by stabilising the troposphere. Such mechanistic differences demonstrate that the climate response to aerosols is unlikely to be a simple mirror of the response to greenhouse gases. As aerosols have
been found to cause greater changes in precipitation (e.g. Richardson et al. (2016); Hodnebrog et al. (2016)) and extreme

events (Samset et al., 2018) per degree of surface temperature change than greenhouse gases, it is important to understand the mechanisms by which regional climate responses to anthropogenic aerosol arise.

Rapid growth in manufacturing and urbanisation, and a reliance on coal for power generation, have resulted in a substantial increase in the emissions of anthropogenic aerosols and their precursors in Asia since the 1950s (Lin et al., 2016). As a result, local air pollution has increased (Tao et al., 2016). It has also been suggested that transported Asian aerosols may account for around a third of background sulphate in the United States (Park et al., 2004). European anthropogenic aerosol (e.g. Aamaas et al. (2016); Liu et al. (2018)), and midlatitude forcing in general (Byrne and Schneider, 2018), have been shown to be more effective at changing climate per unit emission tropical forcing. However, they have been reducing since the 1970s so that Asian aerosols will most likely dominate global trends in coming decades as air quality concerns have the potential to result in large emissions reductions there (van Vuuren et al. (2011); Riahi et al. (2017)).

Only a limited number of studies have focused on the boreal winter (December to February (DJF)) response to Asian anthropogenic aerosol. The majority have focused on boreal summer (June to August (JJA)), in order to examine the aerosol impact on the summer monsoon (Bollasina et al. (2011), Bartlett et al. (2018)) or annual means (e.g. Richardson et al. (2016); Kasoar et al. (2016); Kasoar et al. (2018); Liu et al. (2018); Grandey et al. (2016)), which can make circulation responses difficult to interpret. Liu et al. (2018) and Menon et al. (2002) both identify remote responses to Asian aerosols in JJA. In addition, Chung and Seinfeld (2005), Allen and Sherwood (2011), and Lewinschal et al. (2013) all identify far-field responses to global aerosol emissions in DJF, particularly at high latitudes in the Northern Hemisphere.

A widespread response to Asian aerosol and precursor emissions is typically found in the tropics, where the small Coriolis parameter allows the influence of a regional forcing to spread throughout the tropics (Sobel et al. (2001); Ming and Ramaswamy (2011)). Circulation responses in the Pacific basin that arise as a result of changes in heating near the Equator are likely to be well explained by the models of Matsuno (1966) and Gill (1980). For a heating anomaly on the Equator, such a response will involve low pressure centres on either side of the Equator, with low-level eastward winds along the Equator to the west, and westward winds to the east. Bjerknes (1966) and Bjerknes (1969) present a mechanism for the midlatitude response to a tropical Pacific anomaly, with a heating anomaly leading to a strengthening of the Aleutian low. Extension of the response into the Atlantic basin typically arises via tropically generated Rossby waves (Hoskins and Karoly (1981); Ting et al. (1996); Held et al. (2002); Scaife et al. (2017)). Stratospheric pathways for teleconnections between the tropical Pacific and the North Atlantic have also been proposed (Bell et al. (2009), Allen and Sherwood (2011); Domeisen et al. (2015)).

In this study, we focus on the large-scale temperature and precipitation response to increases in Asian emissions of aerosol and their precursors, 'Asian anthropogenic aerosol' (specifically sulphur dioxide, the precursor of sulphate aerosols, black carbon and organic carbon) since 1980. Asian anthropogenic aerosol has increased rapidly since 1980, and is currently the largest anthropogenic aerosol source region worldwide (Klimont et al., 2013). Our analysis focuses on DJF, as this is the season when atmospheric teleconnections between the Asia-Pacific region and the midlatitudes are strongest (Diaz et al. (2001); Thomson and Vallis (2018)). In Section 3 we examine the response in a fully coupled GCM, before using a reduced-complexity model to help us identify the main drivers of the teleconnection with Europe and North America, which will be discussed in Section 4.

## 2    Method and models

We use two models of different complexity to identify the far-field response to increases in Asian anthropogenic aerosol between 1980 and 2012: a fully coupled atmosphere-ocean general circulation model, HadGEM3-GC2, and a steady-state primitive equation model, Linear University Model of the Atmosphere (LUMA). LUMA only accounts for the direct, stationary response of the atmosphere to a prescribed forcing (heating perturbation). Used alongside HadGEM3-GC2, it provides additional insight into the origins and pathways of the large-scale atmospheric response to Asian anthropogenic aerosols. Such an approach has been successfully used by Teng et al. (2012) and Ming et al. (2011), to demonstrate the potential of Asian emissions of black carbon to influence temperatures in the United States and Northern Hemisphere extratropics respectively, and by Lewinschal et al. (2013) to explain the Northern Hemisphere response to changes in global aerosol emissions.

### 2.1    HadGEM3-GC2

The Met Office Unified Model (Global Coupled configuration 2) HadGEM3-GC2 (Williams et al., 2015) was run with a horizontal resolution of N216 ($\sim$ 60 km) in the atmosphere, and $\frac{1}{4}^{\circ}$ in the ocean. 85 vertical levels are used in the atmosphere. HadGEM3-GC2 uses the same CLASSIC aerosol scheme (Bellouin et al., 2007), as its CMIP5 predecessor, HadGEM2-ES (Collins et al., 2011). Changes in cloud droplet effective radius, for aerosol indirect effects, are a prognostic function of cloud droplet number concentration and cloud liquid water content (Bellouin et al., 2007). HadGEM3-GC2 includes ENDGame (Even Newer Dynamics for Global Atmospheric Modelling of the Environment, Wood et al., 2014), which is a substantial revision of the atmosphere dynamical core relative to the HadGEM2 family of models, and overall shows significant improvements in mean bias and variability compared to previous model configurations (Senior et al., 2016).

The main mean-state biases in HadGEM3-GC2 are described in detail by Williams et al. (2015). Of particular relevance to our study, Williams et al. (2015) find that the excess equatorial easterly wind stress common in earlier versions of the Met Office climate models is improved in HadGEM3-GC2, primarily through a change to the gravity wave drag scheme, which results in El Niño Southern Oscillation and its teleconnections being well simulated.

We compare simulations with time-varying anthropogenic aerosol and precursor emissions against simulations where Asian emissions of anthropogenic aerosols and their precursors (specifically sulphur dioxide, biomass burning aerosol, soot, and fossil fuel organic carbon) were fixed (Figure 1). In the fixed aerosol case, all emissions of anthropogenic aerosol and their precursors are fixed at their 1971-1980 mean values over Asia: (67.5 to 145° E, 5.0 to 47.5° N, illustrated in Figure 2a). The experiments are driven by CMIP5 historical forcings up to and including the year 2005 (Lamarque et al., 2010), and RCP4.5 thereafter (van Vuuren et al. (2011); Meinshausen et al. (2011)).

A historical experiment, with all forcings transient from 1959 to 2012, was performed. This experiment consists of four ensemble members, each initialised from different atmosphere and ocean states. In November 1970, 4 parallel runs with fixed anthropogenic aerosol and precursor emissions (Figure 1a) were branched. We conduct our analysis from 1979 to allow for a residual response from the sudden change in aerosol in 1970. We assume that the response to Asian anthropogenic aerosol alone is given by the difference of these two experiments (historical - fixed Asia). Throughout the paper, we define the response to

the increase in Asian anthropogenic aerosol as the ensemble mean of the difference difference between these two experiments across two periods: (1998-2012)-(1979-1993). The perturbation in aerosol emissions and precursors in this period accounts for half of the Asian increase since 1900. By the 2000s, Asia accounts for 40% of global aerosol and precursor emissions.

## 2.2 LUMA

LUMA is a linearized version of the portable University model of the atmosphere (Fraedrich et al., 1998), which solves the steady-state primitive equations. The equations are linearized around a zonally symmetric basic state, and include tendencies for surface pressure, temperature, divergence, and vorticity (Liakka et al. (2012); Lewinschal et al. (2013)). LUMA has a horizontal resolution of T21 and 10 levels in the vertical.

     The basic state in LUMA is specified from the 1980-2005-mean zonal-mean zonal, meridional, and vertical wind, temper-
ature, and sea level pressure from HadGEM3-GC2. Lewinschal et al. (2013) showed that the anthropogenic aerosol-induced precipitation changes, and the associated diabatic heating, are the primary sources of Rossby waves in response to global aerosol emissions, and that the hemispheric-scale temperature pattern is the result of a wave response. Here, we will examine the role of precipitation anomalies in response to Asian anthropogenic aerosol in driving far-field responses. Precipitation anomalies from HadGEM3-GC2 are multiplied by the latent heat of evaporation to yield a heating rate, which is implemented
in LUMA as a forcing term. The vertical distribution of heating in response to a precipitation anomaly is represented by a Gaussian function that integrates to 1, with a maximum at 550hPa.

     LUMA's linear framework assumes that the full wave response is a linear sum of regionally forced constituents. By forcing LUMA with regional components of the HadGEM3-GC2 response to Asian aerosol emissions, we will identify the main drivers of the full response.

**3   Results**

The increase in Asian anthropogenic aerosol since 1980 results in an increase in aerosol optical depth over Asia (Figure 1b). The ensemble mean increase in sulphate optical depth for (1998-2012)-(1979-1993) is shown in Figure 2(a). It is primarily located in the main emission regions in northern India and eastern China, with some transport over the Indian Ocean due to advection by the winter monsoon circulation. This increase in aerosol optical depth is associated with a distinct local decrease
in cloud top effective radius, which extends into cloudy regions in the West Pacific and the Bay of Bengal (Figure 2(b)). Consistent with other models, aerosol cloud interactions play an important role in the local response to Asian anthropogenic aerosol in HadGEM3-GC2 (e.g. Wang (2015); Chung and Soden (2017); Dong et al. (2019)).

     Changes in cloud-top effective radius can also be seen further afield. While these changes are significant according to a Student's t-test, many of them are not robust across all ensemble members (Figure 6c). Robust changes outside the Asian
region are associated with changes in cloud related to circulation changes that arise as a result of Asian anthropogenic aerosol, rather than via direct modification of the clouds by the aerosols themselves. For example, increased cloud-top effective radius in the north Pacific is due to a decrease in high cloud (Figure 3b) associated with induced circulation changes in the North

Pacific, and the changes in cloud-top effective radius over the Southern Ocean are associated with a shift in the Southern Hemisphere jet (Figure 5d). Such a response has been noted in previous studies, but Southern Hemisphere circulation changes in response to anthropogenic aerosol are generally not robust (Rotstayn (2013); Steptoe et al. (2016); Choi et al. (2018)).

5    The increase in Asian aerosol optical depth, and local increase in cloud albedo via the Twomey effect, is associated with a decrease in downwelling shortwave radiation over both India and eastern China (co-located with the change in aerosol optical depth), and over the tropical West Pacific (Figure 3(a)). The decrease in downwelling shortwave radiation due to the Twomey effect over the West Pacific is enhanced by an increase in the total cloud fraction in the same region (Figure 3(d)). Over the East China Sea, there is an increase in downwelling shortwave at the surface, which could be related to a decrease in local cloud fraction (Figure 3(a,d)). Changes in cloud cover also affect short-wave radiation in areas outside the regions directly affected by 10    aerosol emissions, such as over the southeast Pacific and south Atlantic (Figure 3(a,b,d)). There is a decrease in downwelling longwave radiation (not shown) over the Maritime continent due to moisture convergence (Figure 5d) and associated cloudiness (Figure 3d).

Over China there is no widespread reduction in near-surface temperature (Figure 4(a)), despite the large change in aerosol optical depth there. This is a result of local competition between the direct and indirect effects of the aerosol changes (cf. 15    Wang (2015); Dong et al. (2019)). In fact, there is an increase in temperature east of China (Figure 4(a)), which is related to a decrease in cloud fraction and increase in downwelling shortwave radiation there (Figure 3(a,d)). Such a reduction may be associated with the semi-direct effect.

In contrast to China, the decrease in downwelling shortwave radiation over India results in a widespread decrease in temperature (Figure 3(a), Figure 4(a)), and a local increase in sea level pressure (Figure 5(a)). This causes a southwestward shift in 20    precipitation (Figure 4(b)), from the Bay of Bengal into the western Equatorial Indian Ocean, and draws air from the Maritime continent and west Pacific into the Indian Ocean sector (Figure 5(d)). Related to these changes is a region of descent over India and the Bay of Bengal (Figure 5(c)), and a negative upper-tropospheric geopotential height anomaly over India (Figure 5(b)).

The decrease in downwelling shortwave radiation over the west Pacific causes cooling (Figure 4(a)), and a local increase in sea level pressure (Figure 5(a)). This leads to a strengthening of the westerly monsoon flow over eastern China and southern 25    Japan (Figure 5(d)). The increase in sea level pressure is also related to low level convergence and ascent over the maritime continent (Figure 5(d), Figure 7), which leads to an increase in cloud fraction (Figure 3(d)). Ascent over the Maritime continent is part of a meridional overturning anomaly, which has its descending branch over eastern China and the western tropical North Pacific (Figure 7). This descent results in a reduction in precipitation in the region (Figure 4(b)). The change in meridional overturning circulation in the west Pacific, and the associated cross-equatorial energy flux, acts to moderate some of the 30    interhemispheric top of atmosphere radiation imbalance that results from increases in Asian anthropogenic aerosol.

Negative sea level pressure anomalies and negative 250hPa geopotential height anomalies are located east of Japan, associated with the region of anomalous descent to the east of China (Figure 5(c)). Here, the local circulation changes associated with the increases in Asian emissions interact with the midlatitude jet, initiating a barotropic Rossby wave train that extends around the Northern Hemisphere midlatitudes to Europe, where an anomalous circulation advects cold air from the north east 35    and causes a cold anomaly in Europe (Figure 4(a), Figure 5(a,d)). An anticyclonic anomaly in the North Atlantic also causes

western Europe to be anomalously dry relative to 1979-1993, while the Mediterranean is anomalously wet (Figure 4(b)). This positive North-Atlantic-Oscillation-like response is consistent with the responses seen by Allen and Sherwood (2011) and Westervelt et al. (2018) in response to increases in global aerosols. Note, however, that Westervelt et al. (2018) identified a degree of model-sensitivity in the North Atlantic response to global aerosols, and identified an increase in Mediterranean precipitation that we do not see here.

The extratropical waves seen in Figure 5(a) and (b) have an equivalent barotropic structure. The wave pattern is matched by the pattern of the temperature anomalies, suggesting that the temperature anomalies over the North Atlantic (cool in the subtropics, warm in midlatitudes) and Europe (cool) are the result of adiabatic heating.

In the North Pacific there is a clear weakening of the Aleutian low (Figure 5a), with an associated Equatorward shift in the storm track (Figure 5d), and southeastward shift in precipitation (Figure 4b). This circulation shift is the result of the weakening of the meridional temperature gradient (Figure 4a), and is a characteristic of the Pacific response to anthropogenic aerosol forcing (e.g. Ming and Ramaswamy (2011); Smith et al. (2016)).

In the Equatorial Pacific, the HadGEM3 response to increased Asian anthropogenic aerosol has a La Nina-like pattern, with increased precipitation in the Equatorial west Pacific, and decreased precipitation and anomalous high pressure in the east (Figure 4(b)). Changes further afield, such as the decreased precipitation over western Europe, also resemble characteristic La Nina teleconnections (e.g. increased precipitation in northern South America, southern Africa, and eastern Australia; and a warm anomaly in the southern United States, with a cold anomaly over Alaska). However, other changes, such as the positive-Pacific-North-American-like pattern (Figure 5(a)), with positive geopotential height anomalies over the western United Staes, and negative anomalies to the east, are more typically associated with El-Niño. It is an El-Niño-like stationary wave response, arising primarily as a result of diabatic heating anomalies in the East Pacific, that was identified by Ming and Ramaswamy (2011) in response to global aerosol emissions.

The ensemble mean anomalies discussed in this section are significant at the 10% level. Figure 6 shows that they are also robust across ensemble members, with 3 out of 4 members simulating responses of the same sign in the same region. In the following section, we use LUMA to confirm the causality suggested by HadGEM3-GC2, and identify the main source of the teleconnection between Asia and Europe.

### 3.1 LUMA response to Pacific precipitation anomalies

When initialised with the HadGEM3-GC2 climatology (1980 to 2005 mean), and forced with the global DJF precipitation anomaly field simulated by HadGEM3-GC2 in response to the increase in Asian anthropogenic aerosol, LUMA is able qualitatively to reproduce the general features stationary wave pattern seen in the HadGEM3-GC2 simulation (Figure 8). Note that we do not expect LUMA to reproduce the HadGEM2-GC wave amplitude as this is directly proportional to the magnitude of the forcing in such a linear model (Lewinschal et al., 2013). LUMA captures the negative anomalies east of Japan, over the central North Pacific, and over western Europe. It also captures the positive anomalies over the contiguous United States and the North Atlantic. It simulates a positive anomaly over central Eurasia, although this is displaced to the west relative to HadGEM3-GC2. The wave pattern in LUMA is slightly too zonal compared to HadGEM3-GC2, which is to be expected given

the prescription of a zonally-symmetric base state. There are also notable differences between the two models in the northern high latitudes. The Southern Hemisphere wave pattern simulated by LUMA is also considerably different to the pattern seen in HadGEM3-GC2. This is the result of a higher degree of nonlinearity in the Southern Hemisphere flow. Since we do not expect LUMA to produce an adequate representation of waves in such conditions (Lewinschal et al., 2013), we focus only on the Northern Hemisphere response to Asian anthropogenic aerosol.

In order to determine the main source of the Rossby wave train that leads to large changes in European and North American climate in response to Asian anthropogenic aerosol, we split the Pacific precipitation anomalies into three main components, from which a forcing term is calculated: the La-Nina-like anomaly over the Equatorial east Pacific; the dipole anomaly between China and Indonesia; and the dipole anomaly in the northern North Pacific (indicated in Figure 9 (a), (b), and (d)). As LUMA is a linear model, the sum of the response to these anomalies should account for most of the wave pattern that arises in response to the full global precipitation anomaly field.

The Equatorial Pacific precipitation anomaly drives a Matsuno-Gill-type quadrupole response (e.g. Jin and Hoskins (1995); Kacimi and Khouider (2018)). The response is predominantly confined to the Pacific basin, and the wider tropics, with little impact on Europe or North America (Figure 9(a)).

The dipole precipitation anomaly downstream of the emission region (Figure 4b), between China and Indonesia results in a large dipole atmospheric flow anomaly immediately above the heating region, with a weak wave propagating into the European sector (Figure 9(b)). If we consider only the northern third of this precipitation anomaly (Figure 9(c)), we see that the response shown in Figure 9(b) most likely is a combination of a propagating wave generated to the east of China and south of Japan, and a Matsuno-Gill response generated over the Maritime continent.

While the waves generated immediately downstream of the emission region extend to Europe, this source region alone does not explain the full magnitude or pattern of the total response, particularly over the US and North Atlantic. Most of the amplitude of the waves in these regions is instead a response to the precipitation anomaly in the North Pacific (Figure 9(d)). Figure 9(e) shows that most of the US and North Atlantic stationary wave response induced by changes in Asian anthropogenic aerosol is explained by a combination of the responses to precipitation anomalies between China and Indonesia and over the northern North Pacific. Over the Pacific itself, the Matsuno-Gill response to the Equatorial forcing is necessary to produce the negative anomaly located over the central North Pacific (Figure 9a vs. Figure 9f).

However, a comparison of the response to global anomalies (Figure 9(f)) and the sum of the response to the heating dipoles downstream of China and Indonesia and in the northern North Pacific (Figure 9(e)) shows that these regions alone cannot fully account for the global response. The positive anomaly in the eastern tropical North Atlantic includes some response to anomalies in the Equatorial Pacific (Figure 9(a)), and the magnitude and structure of the wave anomaly over western Europe and Scandinavia are not explained by any of the regions shown in Figure 9(e). Additional simulations exploring the response to Atlantic precipitation anomalies confirm that the structure of the North Atlantic circulation anomalies, and the propagation into western Europe and Scandinavia, are the result of positive feedbacks (Figure 9(g)): precipitation anomalies generated as part of the wave train from the Pacific cause heating in the Atlantic sector, which lead to the wave pattern seen in response to

global precipitation anomalies. Figure 9(h) confirms that it is primarily feedbacks from extratropical precipitation, rather than in the tropical Atlantic, that drive the response.

## 4  Discussion

The pattern of the atmospheric circulation response to increases in Asian anthropogenic aerosols in HadGEM3-GC2 is characteristic of a positive PNA pattern over the midlatitude North Pacific and North America (Figure 4(a), Figure 5(a)). Positive PNA patterns are typically associated with El Niño (Lau (1997); Ming and Ramaswamy (2011)). However, in HadGEM3-GC2, we see such a pattern alongside La-Niña-like temperature and precipitation patterns in the tropical Pacific. This is consistent with our findings that the North American and European responses to Asian anthropogenic aerosol arise primarily in the extratropics. Such a PNA pattern in response to Asian anthropogenic aerosol was also found by Teng et al. (2012), which they also concluded were not tropically excited (note that they considered an increase in Asian black carbon alone, so the response is opposite in sign). Straus and Shukla (2002) found that, although there are superficial similarities between the midlatitude response to ENSO and the PNA, ENSO does not drive the PNA.

Our conclusion that the large-scale circulation response to Asian anthropogenic aerosol is primarily an extratropical-driven phenomenon, either through Rossby wave trains excited in the extratropics, or extratropical meridional temperature gradients, differs from the conclusion of Ming et al. (2011). Using an idealised model, they found that the upper tropospheric circulation response to global aerosols was largest in the Pacific basin, with a south-eastward shift of the Aleutian low. They linked this shift to stationary Rossby waves excited by anomalous diabatic heating over the tropical east Pacific. HadGEM3-GC2 simulates a very different pattern of Equatorial Pacific precipitation changes in response to Asian anthropogenic aerosol to the response to global emissions shown by Ming and Ramaswamy (2011). We find a response that is opposite in sign, and with greater zonal asymmetry. Therefore, the differences between our conclusions are likely to be due to differences in the local response to emissions, rather than differences in the mechanism for the wider midlatitude response to Asian anthropogenic aerosol.

HadGEM3-GC2 has a large bias in meridional wind over the northeast Pacific (Lee et al., 2018), which may affect the propagation of waves into the North American and Atlantic sectors. We tested how this bias in the underlying wind field may influence the stationary wave response by prescribing a NorESM1-M (Iversen et al., 2013) background state in LUMA. NorESM1-M has a different wind climatology, and does not have the North Pacific biases seen in HadGEM3-GC2. The LUMA result is broadly insensitive to the choice of background state (not shown). However, a zonal-mean background wind field is prescribed in LUMA, so such a reduced sensitivity to biases in one basin is to be expected. The ability of LUMA to reproduce the HadGEM3-GC2 wave pattern in this sector, despite the prescription of a zonally symmetric base state also suggests that the strong teleconnections between the Pacific, North America, and Europe seen in HadGEM3-GC2, are not strongly dependent on the features of the background wind field in the model. This insensitivity to changes of the basic state, further points towards the distribution of the thermal forcing as governing the wave propagation.

Heating anomalies prescribed in LUMA are centred at 550hPa, which is appropriate for tropical heating, but is likely to result in relatively too much heating in the free troposphere in the midlatitudes. Repeating the LUMA analysis with the northern

North Pacific heating prescribed at lower levels showed that the LUMA result is largely insensitive to the vertical position of the anomaly (not shown). Teng et al. (2012) also found that remote impacts were not very sensitive to the structure of the heating profile.

HadGEM3-GC2 has a historical aerosol radiative forcing of -2.19 W m$^{-2}$, which is large relative to the CMIP5 ensemble mean of -1.17 W m$^{-2}$ (Zelinka et al., 2014). However, the model is able to skilfully reproduce the observed evolution of global mean temperature (Senior et al., 2016). In addition to a large sensitivity to aerosol emissions, the model also slightly underestimates the amplitude of interannual variability in near-surface temperature. This means that our results are likely to represent an upper bound of the magnitude of the remote climate response to recent increases in Asian emissions. However, this does not affect our conclusions regarding the mechanism for the response.

## 5 Conclusions

Increases in Asian emissions of anthropogenic aerosols and their precursors since 1980 were found to have a global impact in HadGEM3-GC2. Using experiments with a linear baroclinic model, LUMA, in support of the HadGEM3-GC2 results, we show that the radiative effects of Asian anthropogenic aerosol modify the extra-tropical stationary-wave pattern, leading to Northern-hemisphere-wide precipitation and surface-temperature changes.

Asian aerosols cause a warming east of China in winter (DJF), via a decrease in cloud fraction and increase in downwelling shortwave radiation. Changes in temperature and sea level pressure lead to an increase in eastern Maritime continent precipitation via an increase in convergence and ascent. Intensification of the offshore monsoon flow causes a reduction in precipitation east of China. Aerosol increases cause local cooling over India, which is largely a direct effect of the aerosol increase. This cooling causes an increase in sea level pressure and a south-westward precipitation shift.

Remote responses to Asian anthropogenic aerosol increases are associated with a Rossby wave train from the western north Pacific to Europe. Aerosol-induced heating perturbations in both the western and northern North Pacific are important for the structure and amplitude of this wave train, with the response to northern North Pacific forcing being key to the extension of the wave train into Europe. Positive feedbacks with induced precipitation changes over the North Atlantic help to extend the wave train into Scandinavia. Although the increases in Asian anthropogenic aerosol produce a strong La-Nina-like response in the Equatorial Pacific, the dynamical response to this is confined to the tropics, and to the Pacific basin.

Aggressive mitigation of aerosol and precursor emissions may result in them being reduced to their 1950 levels by 2030 (e.g. Scannell et al. (2019)), which is roughly double the magnitude of the perturbation applied here, but opposite in sign. This may result in around half a degree of warming, in addition to that from future increases in greenhouse gas emissions (Hienola et al., 2018). If the teleconnection identified in HadGEM3-GC2 is present in the future, Europe, western Canada, Alaska, and the Arctic, may be particularly sensitive to any rapid climate changes that occur in the near future in response to a reduction of Asian emissions.

*Data availability.* All model data is archived on JASMIN, and can be accessed via request to Laura Wilcox.

*Author contributions.* LJW, EJH, AL, and AE designed the study. ND ran HadGEM3-GC2. AL ran LUMA. LJW performed the analysis. All authors contributed to writing the manuscript.

*Competing interests.* The authors declare that they have no conflict of interest.

5   *Acknowledgements.* This work and its contributors Laura Wilcox, Eleanor Highwood, and Massimo Bollasina were supported by the UK-China Research & Innovation Partnership Fund through the Met Office Climate Science for Service Partnership (CSSP) China as part of the Newton Fund. Laura Wilcox received additional support from the National Centre for Atmospheric Science (NCAS) and International Meteorological Institute (IMI) visiting scientist programs.

The LUMA simulations were performed on resources provided by the Swedish National Infrastructure for Computing (SNIC) at the
10   National Supercomputer Centre (NSC). The analysis in this work was performed on the JASMIN super-data-cluster (Lawrence et al., 2012). JASMIN is managed and delivered by the UK Science and Technology Facilities Council (STFC) Centre for Environmental Data Archival (CEDA).

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

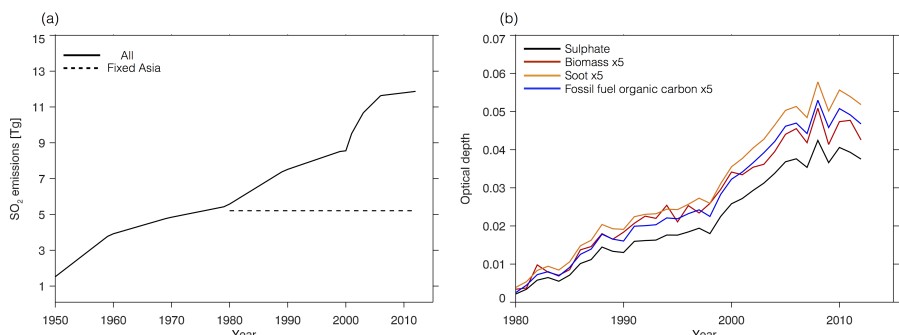

**Figure 1.** (a): Total emissions of sulphur dioxide over Asia from 1950 to 2012 in the historical experiment (solid line). The dashed line indicates 1971-1980 mean values, as used in the fixed Asia experiment. (b): Ensemble mean optical depth of anthropogenic aerosol species in response to Asian aerosol and precursor emissions (historical - fixed Asia), averaged over Asia.

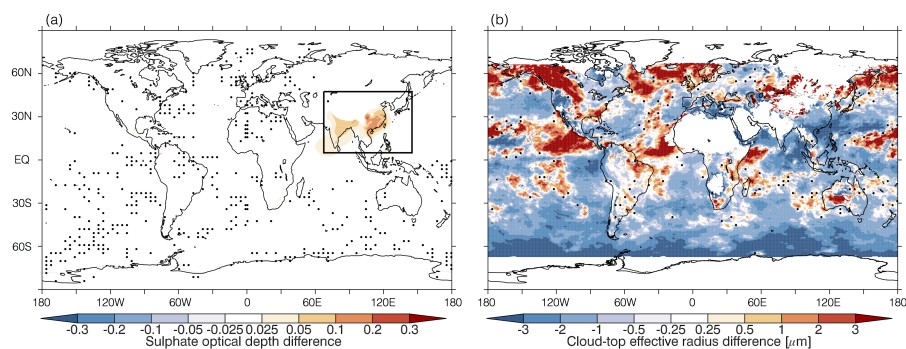

**Figure 2.** Ensemble mean (1998-2012)-(1979-1993) difference in (a): sulphate optical depth (with the black box indicating the emission region), and (b): cloud top effective radius. Dots indicate where the response is not significant at the 10% level, as calculated using a two-tailed Student's t-test. Figure 6c shows the agreement of individual ensemble members on the sign of the anomaly in cloud-top effective radius shown in panel b.

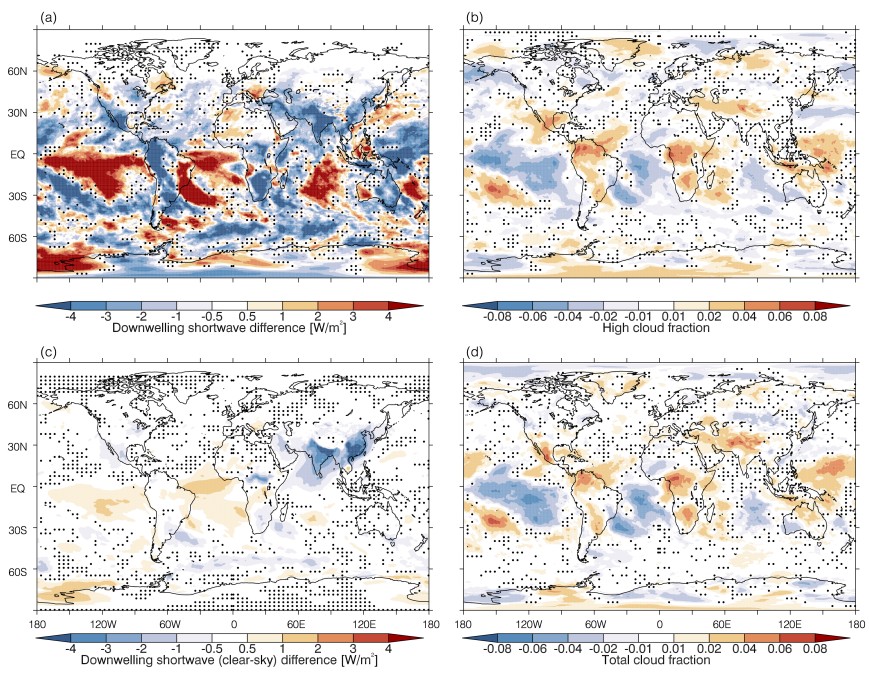

**Figure 3.** Ensemble mean (1998-2012)-(1979-1993) difference in (a): downwelling shortwave at the surface, (b): high cloud fraction, (c): downwelling shortwave at the surface (clear-sky), and (d): total cloud fraction. Dots indicate where the response is not significant at the 10% level, as calculated using a two-tailed Student's t-test.

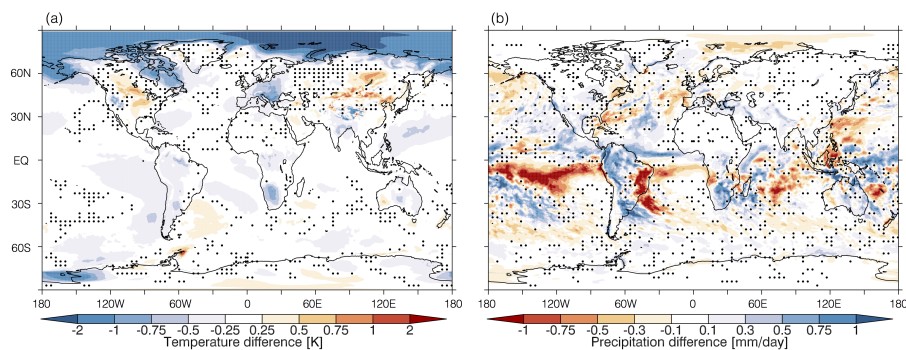

**Figure 4.** Ensemble mean (1998-2012)-(1979-1993) difference in (a): near-surface temperature, and (b): precipitation. Dots indicate where the response is not significant at the 10% level, as calculated using a two-tailed Student's t-test. Figure 6a and b show the agreement of individual ensemble members on the sign of the temperature and precipitation anomalies respectively.

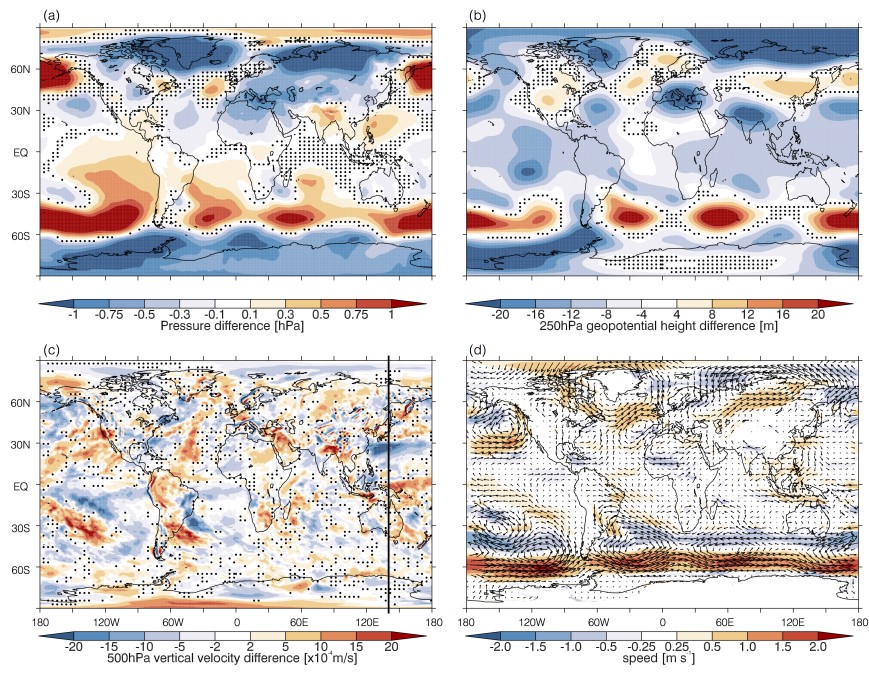

**Figure 5.** Ensemble mean (1998-2012)-(1979-1993) difference in (a): sea level pressure, (b): 250hPa geopotential height, (c): 500hPa vertical velocity, and (d): 850hPa wind. Dots indicate where the response is not significant at the 10% level, as calculated using a two-tailed Student's t-test (note that there is no indication of significance in panel (d)). Figure 6a and b show the agreement of individual ensemble members on the sign of the geopotential height anomaly shown in panel b. The black line in panel (c) is at 140°E, and indicates the location of the transect shown in in Figure 7.

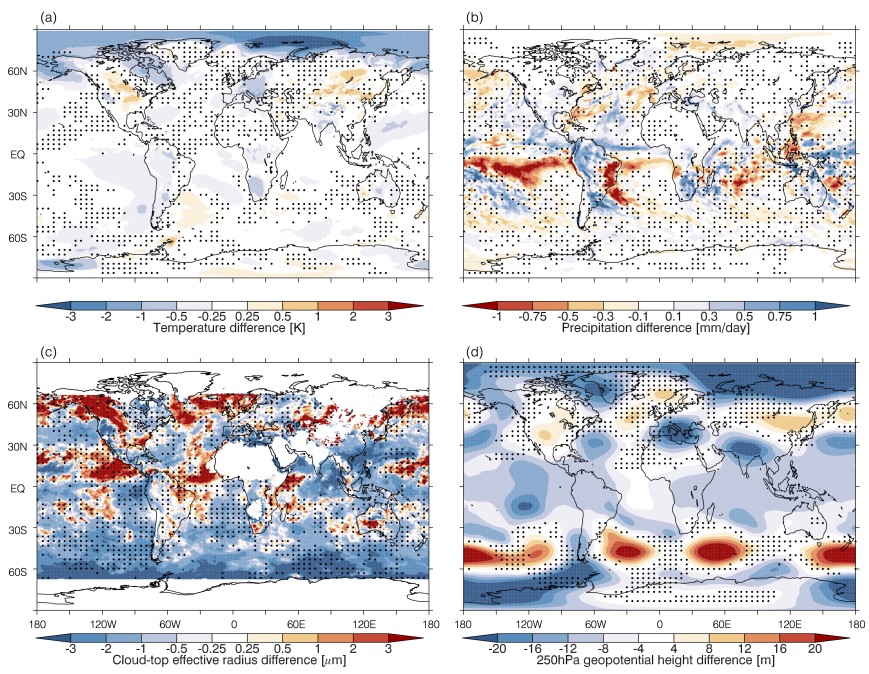

**Figure 6.** Ensemble mean (1998-2012)-(1979-1993) difference in (a): near-surface temperature, (b): precipitation, (c): cloud-top effective radius, and (d): 250hPa geopotential height. In regions without stippling, the anomaly has the same sign in three or more ensemble members.

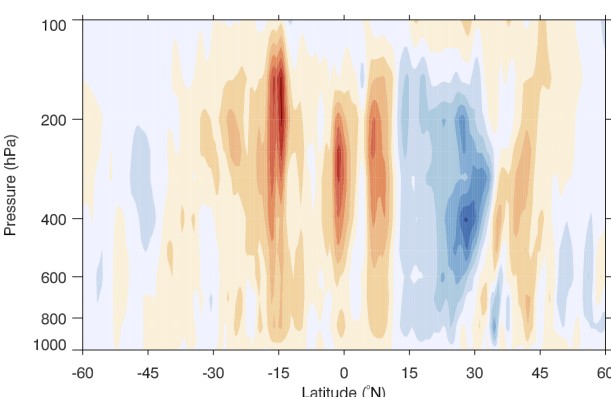

**Figure 7.** Vertical velocity [m s$^{-1}$] averaged over the 5 degrees of longitude centred on 140°E. Positive values indicate ascent.

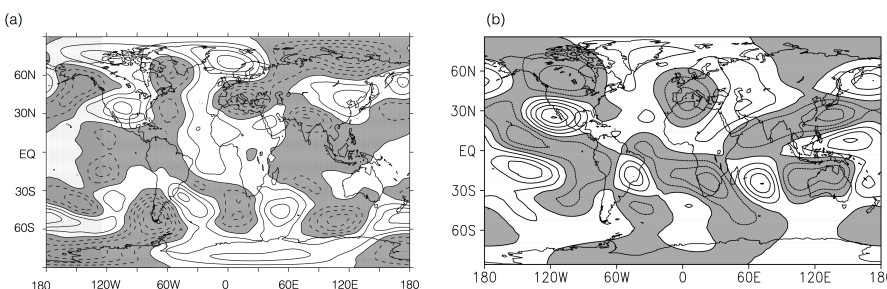

**Figure 8.** (a): (1998-2012)-(1979-1993) difference in 250hPa stationary waves in HadGEM3-GC2. (b): 250hPa stationary wave pattern from LUMA when driven by the global HadGEM3-GC2 precipitation anomaly. In both panels, grey shading/dashed contours indicate negative anomalies, white shading/solid contours indicate positive anomalies.

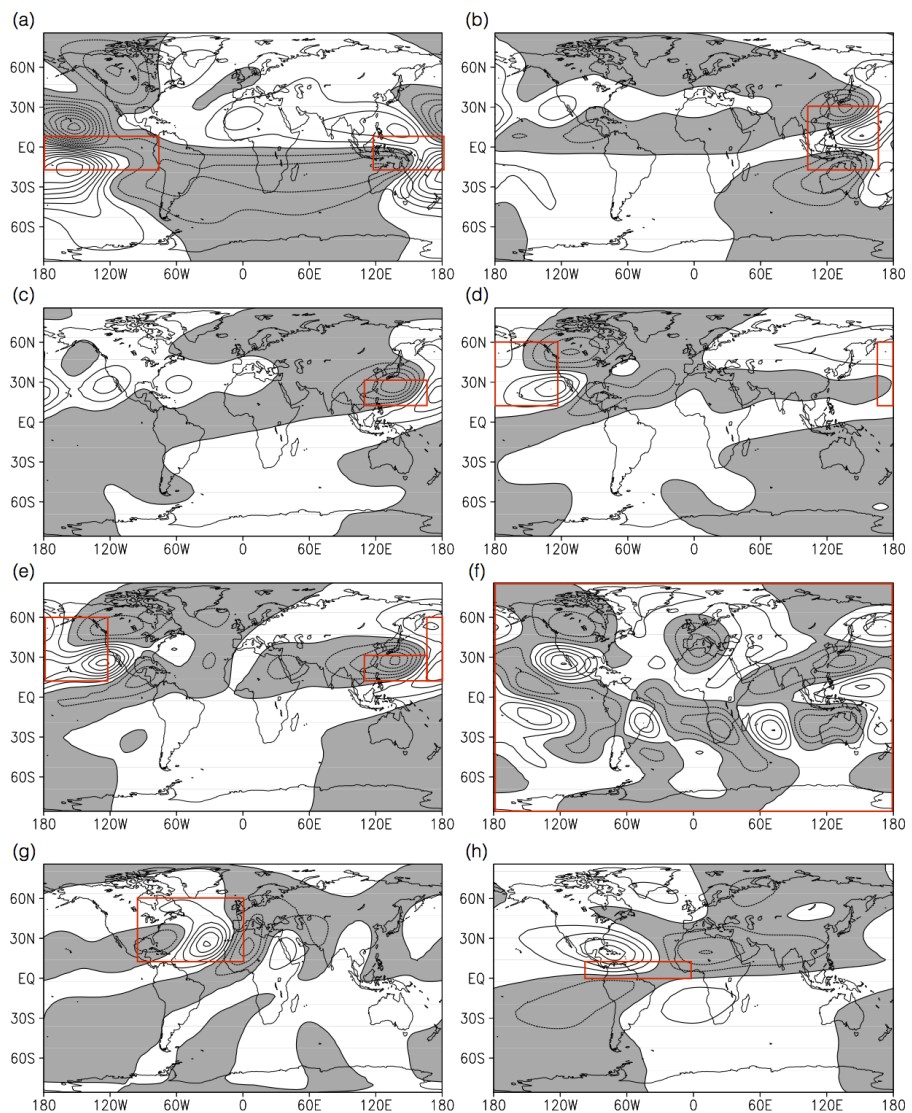

**Figure 9.** 250hPa stationary wave anomaly from LUMA when forced with the HadGEM3-GC2 (1998-2012)-(1979-1993) precipitation anomaly from (a): the Equatorial Pacific [120° E:75° W, 15° S:10° N]; (b): downstream of China and Indonesia [100:160° E, 15° S:30° N]; (c): downstream of China [110:160° E, 15:30° N]; (d): the northern North Pacific [160:240° E, 15:60° N]; (e): the sum of regions (c) and (d); (f): the whole globe; (g): the North Atlantic [15:60° N, 0:90° W]; and (h): the tropical North Atlantic [0:15° N, 0:90° W]. Grey shading/dashed contours indicate negative anomalies, white shading/solid contours indicate positive anomalies. Red boxes indicate the location of the precipitation anomaly.