# Peer review of "Mechanisms for a remote response to Asian anthropogenic aerosol in boreal winter"

_Atmospheric Chemistry and Physics, 2018_

## Referee Comment (RC1) · Anonymous Referee #3 · 28 Dec 2018

The goal of this study is to understand the mechanisms by which changes in anthropogenic aerosol and precursor emissions from Asia drive regional responses in temperature and precipitation during boreal winter. The study uses a combination of a fully coupled atmosphere-ocean general circulation model and a steady-state primitive equation model. The topic is of interest to readers of Atmospheric Chemistry and Physics. There are some interesting and informative results in the study; however, some aspects of the analysis in the current version of the manuscript is a bit cursory. I recommend semi-major revisions to the manuscript before publication.

**Specific Comments**

Section 2 should provide a summary of emissions used in the study, including showing timeseries plots of 1975-2007 emissions from Asia of key species. I am assuming SO2,

[Figure]

NOx, CO, black carbon, and organic carbon emissions were included, but this should be made clear and explicit.

The analysis shown the manuscript is quite cursory. For example, the study does not discuss the difference between aerosol scattering and absorption and how the difference between scattering and absorption may have contributed to the differences in local responses to emissions changes. The distinctions between the impacts of aerosol-radiation interactions vs. those of aerosol-cloud interactions were also not discussed. It may be useful to use the method of Ghan (2013) to calculate direct radiative effect and indirect effect as a way to tease out how different effects of aerosols and differences in spatial distributions of different forcings impact local- and large-scale changes.

The manuscript is unclear on what HadGEM3-GC2 simulation setups were used and how they were analyzed to determine the changes due to anthropogenic aerosol and precursor emissions from Asia. Section 2.1 states "We compare simulations with time-varying anthropogenic aerosol emissions (1975-2007) against simulations where Asian anthropogenic emissions were fixed [at their 1970-1981 mean values]", suggesting that there were two sets of 1975-2007 simulations and that the difference between the two sets were taken as the response to Asian emissions changes. However, the same paragraph also states "Throughout the paper, we define the response to the increase in Asian emissions as the difference between two periods: (1993-2007)-(1979-1993)", suggesting that there was only one set of simulations and that the response was taken as the difference of two time periods from the same set of simulations.

In the captions of Figures 1 to 4 does "not significant at the 10% level" mean "not significant at the 90% confidence level"? The manuscript should indicate how the confidence levels were calculated.

Figure 1b shows that the regions with decrease in cloud top effective radius extend much beyond the West Pacific and the Bay Bengal noted in the text (lines 18-19 of

page 5). What explains such a large extent of decrease in cloud top effective radius in response to emission changes only in Asia? Rather than or in additional to cloud fraction (Figure 2d), it may be useful to examine cloud optical depth.

". . .decrease in downwelling shortwave radiation over both India and China" in lines 22-23 of page 5 is mis-leading as Figure 2a shows an increase in downwelling shortwave radiation in large parts of China.

Lines 26 and 32 of page 5: Could the slight decrease in cloud fraction in eastern China be due to semi-direct effect?

Line 2 on page 6: By "southwesterly shift", do you mean "southwestward shift"? Southwesterly means coming from the southwest whereas Figure 3b suggests precipitation zone shifting towards the southwest.

One of the maps in Figure 4 should have 130E labeled (i.e., the longitude shown in Figure 5).

**Technical Corrections**

- Throughout, "aerosol emissions" should be "aerosol and precursor emissions".

- Page 1, Line 3 (Abstract): For clarity, suggest revising ". . .to isolate the impact of Asian aerosols on global climate. In boreal winter, it is found. . ." to ". . .to isolate the impact of aerosol and precursors emissions from Asia on global climate during boreal winter. It is found. . ."

- Page 1, Line 9 (Abstract): The meaning of "positive" in "positive-Pacific-North-American circulation pattern is not clear here.

- Page 1, Line 18: "provide additional" → "can act"

- Page 2, Line 7: "of the order of weeks" should be "of orders of a few days to a couple of weeks"

- Page 2, Line 7: "heterogeneous" → "spatially heterogeneous"

- Page 2, Line 19: The meaning of the first sentence of the paragraph is unclear, suggest changing it to "Some studies have shown that the spatial patterns of temperature and precipitation responses are similar regardless of the regional locations of the aerosol and precursor emission changes..."

- Page 3, Line 3: "air quality as declined" → "air pollution has increased"

- Page 3, Line 25: Suggest having "In this study..." be the start of a new paragraph.

- Page 4, Line 26: Explain how the four ensemble members are different.

- Figure 1a should include a box indicating where emissions are considered to be in Asia in the HadGEM3-GC2 simulations.

**References**

Ghan, S. J.: Technical Note: Estimating aerosol effects on cloud radiative forcing, Atmos. Chem. Phys., 13, 9971–9974, https://doi.org/10.5194/acp-13-9971-2013, 2013.

---

## Referee Comment (RC2) · Anonymous Referee #1 · 3 Jan 2019

The authors present an investigation into the mechanisms behind the climate responses to Asian aerosol emission, based on two quite different climate models. Using the HadGEM3 fully coupled model they simulate a global climate evolution forced by historical and RCP4.5 emission (four member ensemble), and one where Asian aerosol emissions are kept constant (also four ensembles). The results are interpreted separately, and also used as input to the LUMA steady-state model.

The topic of the paper is timely and interesting, and the tools well suited for the investigation. I like what the authors are trying to do, and believe that the paper can make a strong contribution to the field. However, as it stands, there are some key pieces of information missing that make it difficult to judge how much weight can be put on the detailed results in the second half of the paper. (See below.) Once these are added,

and assuming the results hold up, I recommend that the paper be published in ACP.

Comments:

From the present manuscript, it is difficult to know the strength of the perturbation applied, and therefore what amount of signal should be expected. The AOD in Figure 1a gives an indication, but as the authors themselves state there are so many compensating effects and nonlinearities in this region that AOD does not directly translate into a forcing. I would recommend adding 2-3 simple fixed SST calculations to estimate the ERF, e.g. using the mean emissions of the two periods used (1993-2007 and 1979-1993) and the fixed aerosol emission case. This will greatly aid the reader in interpreting the results.

All figures are presented as the difference (1993-2007) - (1979-1993), presumably as means of the four fully coupled ensemble members? It would be good to see some plots of the individual members too, to get a feel for internal variability. And how is significance calculated? This is crucial. Should we e.g. really believe that the small dAOD shown in Figure 1a causes the large and significant (to 10%) change in cloud top effective radius over western Canada in Figure 1b?

Another critical question is how the fixed aerosol case was spun up? And are they of equal length to the transient runs? I assume the transient simulations branch off from a historical run, but if the aerosol emissions are suddenly set to the 1970-1981 mean at the same time then there will be a residual response during the first years. (This is probably not what was done, but the mehtodology isn't currently described.) Also, is the surface temperature of the fixed emission run consistent with the mean point of the transient runs?

Finally, can the authors use their results to discuss the climate implications of the current strong reductions in (some) Asian aerosol emission sources? This would add an extra layer of relevance to the paper.

(I have no minor comments; the paper is really very clearly written and seems thoroughly proofread.)

---

## Referee Comment (RC3) · Anonymous Referee #2 · 4 Jan 2019

The topic is interesting to ACP readers and the paper includes useful and informative results. I would recommend that the paper is published in ACP after addressing the following comments:

Section 2.1 states "We compare simulations with time-varying anthropogenic aerosol emissions (1975-2007) against simulations where Asian anthropogenic emissions were fixed. Is it possible to conduct simulations for a longer time period for example 1975-2015?

Figure 1a should include a box showing where the sources for emissions are considered, and the color bar should show negative and positive values clearly for all figures, for example in Figure 1a many areas are white, which is not possible to distinguish the negative and positive values.

[Figure]

Figure 1b shows significant negative values around 60 degrees South. The paper should explain the reason for these negative values.

Fig 3b shows (1993-2007)-(1979-1993) difference in precipitation, again it is hard to distinguish negative and positive values and compared to Liu et al. (2018) Fig 1, these values are much smaller and the pattern is not clear. Of course, they look at annual values and this paper focuses on boreal winter but it is needed to look at the annual values too and compare it with PDRMIP models because this paper uses only one model.

Fig6 compares HadGEM3-GC2 with LUMA qualitatively only, it should also include values.

---

## Author Comment (AC1) · 18 Feb 2019

We thank the reviewers for their constructive comments. In our response, referee comments are indicated in **bold**, with our comments and changes to the manuscript in plain text. In addressing the reviewers' comments, we have added two new figures to the manuscript. Throughout our response, when discussing figures, we give both the original and revised figure number.

**Anonymous Referee 1**

**From the present manuscript, it is difficult to know the strength of the perturbation applied, and therefore what amount of signal should be expected. The AOD in Figure1a gives an indication, but as the authors themselves state there are so**

**many compensating effects and nonlinearities in this region that AOD does not directly translate into a forcing. I would recommend adding 2-3 simple fixed SST calculations to estimate the ERF, e.g. using the mean emissions of the two periods used (1993-2007 and1979-1993) and the fixed aerosol emission case. This will greatly aid the reader in interpreting the results.**

Unfortunately, we have not been able to perform these additional experiments. We have added additional text on line 2 of page 5 that puts the emission perturbation applied in our experiments in the context of global and Asian historical emissions. We have also added a new Figure 1a, which shows Asian sulphur dioxide emissions since 1950 to provide further information on the size of our perturbation. We hope that such information will help in aiding the reader with the interpretation of our results.

**All figures are presented as the difference (1993-2007) - (1979-1993), presumably as means of the four fully coupled ensemble members? It would be good to see some plots of the individual members too, to get a feel for internal variability. And how is significance calculated? This is crucial. Should we e.g. really believe that the small dAOD shown in Figure 1a causes the large and significant (to 10%) change in cloud top effective radius over western Canada in Figure 1b?**

All our HadGEM3-GC2 figures show the ensemble mean anomaly, and we now state this in the captions. The indicators of significance (or lack thereof) are based on a Student's t-test, and we have now specified this in the figure captions in the revised manuscript.

We agree that the behaviour of the individual ensemble members is interesting. Instead of using 4 panels to show the response from the 4 members for a single variable, we've instead included a 4 panel figure showing ensemble consensus for 4 of the variables shown in other figures: cloud top effective radius, near-surface temperature, precipitation, and 250hPa geopotential height. This is Figure 6 in the revised manuscript. This enables the reader to get a sense of the internal variability, but also provides another

means by which to assess the robustness of our results. We have added additional text relating to this new figure beginning at line 21 on page 7, and in the captions of Figures 2, 4, and 5.

The example of Canadian effective radius is an interesting one. We do believe that this is caused by the dAOD show in Figure 1a, but consider it unlikely to be an indication that Asian aerosol is directly affecting the clouds in this region. What we think we're seeing here is a change in the clouds as a result of the circulation changes induced over the Aleutian Islands and Canada. We've tried to make this clearer in the text at line 28 of page 5, as we agree that such large changes in effective radius so far from the aerosol emission region are very striking. We have also replaced the downwelling longwave panel in our original Figure 2 (revised Figure 3) with a panel showing the change in high cloud fraction to illustrate this mechanism. We have also included a panel in the new Figure 6, which shows that many of the remote cloud-top effect radius changes are not robust across ensemble members (although the Canadian change is robust).

**Another critical question is how the fixed aerosol case was spun up? And are they of equal length to the transient runs? I assume the transient simulations branch off from a historical run, but if the aerosol emissions are suddenly set to the 1970-1981 mean at the same time then there will be a residual response during the first years. (This is probably not what was done, but the mehtodology isn't currently described.) Also, is the surface temperature of the fixed emission run consistent with the mean point of the transient runs?**

We have added this additional detail to our methodology beginning at line 24 on page 4.

**Finally, can the authors use their results to discuss the climate implications of the current strong reductions in (some) Asian aerosol emission sources? This would add an extra layer of relevance to the paper.**

We have added some additional speculation on future changes to from line 24 on page 10. However, there are many uncertainties associated with the climate response to current (and potential future) reductions in Asian aerosol emissions, which mean that we need to be cautious in using our historical results to speculate in this way. A recent study by Hienola et al. (now cited in the manuscript) suggests that future reductions in aerosol emissions (occurring primarily over Asia) may result in an additional 0.5C of global warming by 2050, in addition to that due to greenhouse gas increases.

**Anonymous Referee 2**

**Section 2.1 states "We compare simulations with time-varying anthropogenic aerosol emissions (1975-2007) against simulations where Asian anthropogenic emissions were fixed. Is it possible to conduct simulations for a longer time period for example1975-2015?**

We have extended our HadGEM3-GC2 simulations to 2012, and have rerun our LUMA simulations with an appropriately revised precipitation anomaly. We now present figures showing the response to increases in Asian emissions between (1998:2012)-(1975:1989). All figures have been revised to reflect this extension, but there is no material change in our results.

**Figure 1a should include a box showing where the sources for emissions are considered, and the color bar should show negative and positive values clearly for all figures, for example in Figure 1a many areas are white, which is not possible to distinguish the negative and positive values.**

We have added a box to Figure 1a (revised Figure 2a) showing where we have perturbed our emissions.

Having a white central value in a divergent colour bar makes it easier to see positive and negative values. It allows the reader to clearly see the large responses, without the distraction of additional colours for insignificant components of the response. In

Figures 2-4, where we are looking at the response to emissions, the white regions are very closely matched by our stippling that indicates a lack of significance at the 10% level. Figure 1a is an unusual case, with lots of white space as we don't expect to see any change in most areas of the world since we are only perturbing emissions in Asia. We considered showing the AOD change in Figure 1a over a reduced spatial domain to reduce the amount of white space, but think it is better to show a global domain for ease of comparison to the other figures in the manuscript, and to confirm that the only changes in anthropogenic aerosol optical depth in our experiments are over Asia.

**Figure 1b shows significant negative values around 60 degrees South. The paper should explain the reason for these negative values.**

The negative values around 60S are associated with a shift in the Southern Hemisphere jet, and the associated change in cloud. This circulation shift can be seen in our original Figure 4 (revised Figure 5). There have been a number of mechanisms proposed to link changes in the Southern Hemisphere circulation to predominantly Northern Hemisphere aerosols, but this is not a response that is robust across models. We've noted this in the text, beginning at line 31 on page 5, and pointed to some relevant literature.

**Fig 3b shows (1993-2007)-(1979-1993) difference in precipitation, again it is hard to distinguish negative and positive values and compared to Liu et al. (2018) Fig 1, these values are much smaller and the pattern is not clear. Of course, they look at annual values and this paper focuses on boreal winter but it is needed to look at the annual values too and compare it with PDRMIP models because this paper uses only one model.**

Our checks show that our Figure 3b (revised Figure 4b) is accessible to red-green and blue-yellow colour blind readers. It should be just as easy for most readers to distinguish between our reds and blues as between the browns and blues used by Liu et al. (2018).

The values shown in our Figure 3b are indeed much smaller than those in Figure 1 of Liu et al. (2018). This is partly due to our use of mm/day compared to their mm/year. However, the difference is primarily the result of the very different experimental design used in PDRMIP (as analysed by Liu et al.) compared to our experiment. The regional aerosol perturbation experiments shown in Figure 1 of Liu et al. (2018) are the difference between present day and a 10x scaling of present-day aerosol emissions (or concentrations, depending on the model) in the relevant region. In our work, we consider the difference between realistic present day emissions, and 1980s emissions (roughly half of the present day values). In addition to using a larger emission perturbation, Liu et al. (2018) also consider the equilibrium response, which one might also expect to be larger than the response in the transient simulations we consider here.

Although there is a lot to be gained from studying the annual mean response, it would be inappropriate to do so here. We are interested in the dynamical response to aerosol, the features of which are seasonally dependent. Similarly, looking at PDRMIP here would substantially change the character of our study, and complicate the results by introducing a very different experimental design. The dynamical response in the PDRMIP models is likely to be affected by the different mean state biases in the models. A multi-model ensemble mean is therefore likely to obscure some of the dynamical mechanisms, and analysis of the responses in the individual models will mean dedicating a large portion of the manuscript to explaining any differences between the model mean states and their responses to aerosol. This is something that we intend to explore in PDRMIP in the future. However, we think it would detract from the novel analysis of models of different complexities we have presented to include such analysis in this manuscript.

**Fig6 compares HadGEM3-GC2 with LUMA qualitatively only, it should also include values.**

As explained in the original text at line 12, page 7, we would not expect LUMA to capture the magnitude of the waves seen in GC2. To avoid confusion, we think it

is better not to show the values for either model in Figures 6 and 7. Quantitative information on the dynamical response is provided in Figure 4 (revised Figure 5). The text explaining why the comparison between HadGEM3-GC2 and LUMA is qualitative is found at line 28, page 7, in the revised manuscript.

**Anonymous Referee 3**

**Section 2 should provide a summary of emissions used in the study, including showing timeseries plots of 1975-2007 emissions from Asia of key species. I am assuming SO2, NOx, CO, black carbon, and organic carbon emissions were included, but this should be made clear and explicit.**

We have included a new Figure 1a showing timeseries of emissions of sulphur dioxide over Asia to better illustrate our experimental design. We now explicitly list all the species we have perturbed in our simulations in the methodology on line 24, page 4, and show timeseries of the optical depth of all anthropogenic aerosol species in Figure 1b.

**The analysis shown the manuscript is quite cursory. For example, the study does not discuss the difference between aerosol scattering and absorption and how the difference between scattering and absorption may have contributed to the differences in local responses to emissions changes. The distinctions between the impacts of aerosol-radiation interactions vs. those of aerosol-cloud interactions were also not discussed. It may be useful to use the method of Ghan (2013) to calculate direct radiative effect and indirect effect as a way to tease out how different effects of aerosols and differences in spatial distributions of different forcings impact local- and large-scale changes.**

We agree that the processes underlying the local responses to emissions changes are interesting. However, we wanted the focus of this paper to be on the dynamical mechanisms underlying the remote response to Asian emissions. There is a lot going on locally, and it would detract from the analysis of both components of the analysis to

try to cover them in detail in one manuscript.

Thank you for directing us to Ghan (2013). Unfortunately, we didn't archive the necessary variables to apply this method in our work. However, we have run additional simulations where we turn off aerosol indirect effects in HadGEM3 in order to explicitly examine the different effects of aerosols. We find that the magnitude of the local forcing and response is dominated by aerosol indirect effects in this model. We also use additional prescribed sea surface temperature (SST) experiments to demonstrate the importance of aerosol-induced SST changes to the local dynamical response to aerosol forcing. These results should be appearing soon in Climate Dynamics: Dong et al., "Impacts of recent decadal changes in Asian aerosols on the East Asian summer monsoon: roles of aerosol-radiation and aerosol-cloud interactions", which we have cited in the results section of this manuscript.

**The manuscript is unclear on what HadGEM3-GC2 simulation setups were used and how they were analyzed to determine the changes due to anthropogenic aerosol and precursor emissions from Asia. Section 2.1 states "We compare simulations with time-varying anthropogenic aerosol emissions (1975-2007) against simulations where Asian anthropogenic emissions were fixed [at their 1970-1981 mean values]", suggesting that there were two sets of 1975-2007 simulations and that the difference between the two sets were taken as the response to Asian emissions changes. However, the same paragraph also states "Throughout the paper, we define the response to the increase in Asian emissions as the difference between two periods: (1993-2007)-(1979-1993)",suggesting that there was only one set of simulations and that the response was taken as the difference of two time periods from the same set of simulations.**

We have expanded our methods section to include a more detailed description of both our experimental design and our approach to the analysis in the paragraphs beginning on lines 23 and 29 of page 4.

**In the captions of Figures 1 to 4 does "not significant at the 10% level" mean "not significant at the 90% confidence level"? The manuscript should indicate how the confidence levels were calculated.**

Results that are not significant at the 10% level are those that are found within the 90% confidence interval. The significance level is the probability of rejecting our null hypothesis (that there is no difference between (1993-2007)-(1979-1993) in response to Asian emissions) when it is true. We have clarified our approach to significance testing in the revised figure captions.

**Figure 1b shows that the regions with decrease in cloud top effective radius extend much beyond the West Pacific and the Bay Bengal noted in the text (lines 18-19 of page 5). What explains such a large extent of decrease in cloud top effective radius in response to emission changes only in Asia? Rather than or in additional to cloud fraction (Figure 2d), it may be useful to examine cloud optical depth.**

We consider it unlikely that the changes in these remote regions are an indication that Asian aerosol is directly affecting the clouds so far from the emission region. What we think we're seeing here is a change in the clouds as a result of the circulation changes induced over the Aleutian Islands and Canada. We've tried to make this clearer in the text, beginning at line 28 on page 5. We don't have cloud optical depth available. Instead, we have added a panel to Figure 3, which illustrates the high cloud changes as part of this mechanism. We have also added an additional figure, revised Figure 6, which shows the robustness of the cloud-top effective radius changes across our four ensemble members.

**"...decrease in downwelling shortwave radiation over both India and China" in lines 22-23 of page 5 is misleading as Figure 2a shows an increase in down-welling shortwave radiation in large parts of China.**

We have amended this sentence to specify that the changes over China are co-located

with the change in aerosol optical depth. It is now found starting on line 5 of page 6.

**Lines 26 and 32 of page 5: Could the slight decrease in cloud fraction in eastern China be due to semi-direct effect?**

Yes. We've noted this possibility in the text at line 16 of page 6.

**Line 2 on page 6: By "southwesterly shift", do you mean "southwestward shift"? South-westerly means coming from the southwest whereas Figure 3b suggests precipitation zone shifting towards the southwest.**

Yes, thank you for catching this. We have amended the text, which is now a line 18 on page 6.

**One of the maps in Figure 4 should have 130E labeled (i.e., the longitude shown in Figure 5).**

Agreed. We've added this to Figure 5c (original Figure 4c), and link it to Figure 7 (original Figure 5) in the figure captions. Note that we now show a transect at 140E to better reflect the structure of the circulation anomaly in the extended HadGEM3-GC2 simulations used in the revised manuscript.

**Technical Corrections**

**Throughout, "aerosol emissions" should be "aerosol and precursor emissions".**

Done

**Page 1, Line 3 (Abstract): For clarity, suggest revising "...to isolate the impact of Asian aerosols on global climate. In boreal winter, it is found..." to "...to isolate the impact of aerosol and precursors emissions from Asia on global climate during boreal winter. It is found..."**

Done

**Page 1, Line 9 (Abstract): The meaning of "positive" in "positive-Pacific-North-**

[Figure]

**American circulation pattern is not clear here.**

The positive phase of the PNA is associated with above normal geopotential heights over the western US and below normal geopotential heights over the eastern US. Since this is a widely recognised teleconnection pattern it didn't feel appropriate to define it in the abstract. Instead we have added a description of the pattern in the text at line 17 of page 7.

**Page 1, Line 18: "provide additional"→"can act"**

Done

**Page 2, Line 7: "of the order of weeks" should be "of orders of a few days to a couple of weeks"**

Done

**Page 2, Line 7: "heterogeneous"→"spatially heterogeneous"**

Done

**Page 2, Line 19: The meaning of the first sentence of the paragraph is unclear, suggest changing it to "Some studies have shown that the spatial patterns of temperature and precipitation responses are similar regardless of the regional locations of the aerosol and precursor emission changes..."**

Done

**Page 3, Line 3: "air quality as declined"→"air pollution has increased"**

Done

**Page 3, Line 25: Suggest having "In this study..." be the start of a new paragraph.**

Done

**Page 4, Line 26: Explain how the four ensemble members are different.**

Done

**Figure 1a should include a box indicating where emissions are considered to be in Asia in the HadGEM3-GC2 simulations.**

Done

---

## Author Response (AR2)

We are grateful for your careful and constructive comments. In our response, referee comments are indicated in **bold**, with our comments and changes to the manuscript in plain text. We have also corrected a few typos, which can be seen in the marked up version of the manuscript.

**It was suggested in the last around of review that the phrase 'aerosol emissions' should be replaced with 'aerosol and precursor emissions'; however, there are still a few places in the text where 'Asian aerosol emissions' is used in an inappropriate way. Most of the aerosol mass does not come from direct emissions, but rather is formed by chemical processes involving gas-phase species, e.g., sulfate from oxidation of sulphur dioxide; nitrate from nitric acid, which is formed from chemical reactions involving NOx; etc. Thus, 'aerosol emissions' account for a small portion of the total aerosol mass (and small portion of aerosol optical depth). Where using 'Asian aerosol and precursor emissions' is cumbersome, using 'Asian anthropogenic emissions' would be more appropriate than 'Asian aerosol emissions'.**

We agree that it is both the emissions of aerosols and their precursors that have been perturbed in our experiments. We want to avoid referring to 'Asian anthropogenic emissions' as this could also be misleading since we don't perturb greenhouse gases. As a compromise between clarity and readability, we now refer to 'Asian anthropogenic aerosol' throughout the central sections of the manuscript, and define this usage on line 29 of page 3. We have kept the full 'Asian emissions of anthropogenic aerosols and their precursors' in the abstract and conclusions so that there is no ambiguity in these sections, which may be read in isolation. We have also changed the title of the paper.

**Page 3, line 31: What about increases in nitrogen oxides (NOx = NO + NO2) and ammonia emissions? NOx contributes to aerosol nitrate and ammonia contributes to aerosol ammonium.**

The CLASSIC aerosol scheme, which we use in our version of HadGEM3, contains numerical representation of up to eight tropospheric aerosol species: ammonium sulphate, mineral dust, sea salt, fossil fuel black carbon, fossil fuel organic carbon, biomass burning aerosols, secondary organic (also called biogenic), and ammonium nitrate aerosols. Nitrate aerosol was not included in either the main CMIP5 simulations with HadGEM2-ES (see Bellouin et al., 2011, for example), or in our simulations with HadGEM3-GC2, as the nitrate aerosol scheme was still in development at the time the simulations were run.

**Page 4, line 28: Does 'biogenic aerosol' mean bioaerosol or is this an error? Other than bioaerosols, there is no biogenic aerosols that is emitted. Rather, biogenic aerosol typically refers to biogenic secondary organic aerosol (SOA) that are formed in the atmosphere from oxidization of biogenic volatile organic compounds (VOCs). Regardless, ?biogenic? by definition is not anthropogenic, making the mention of 'biogenic' in this sentence about anthropogenic emissions confusing. Collins et al. (2011) indicate that HadGEM2 uses climatology for biogenic aerosols, suggesting that year-specific biogenic emissions are not used for the purpose of biogenic aerosols in HadGEM. These contradictions need to be clarified.**

We agree that biogenic aerosol typically refers to secondary organic aerosol. It is an error to include it here and we have removed it, thank you.

**Figure 1b: Similar to the above comment, having 'Biogenic x5' here in a figure about anthropogenic contribution is confusing. Even more confusing is why the biogenic contribution has an increasing trend that closely follows the trend of anthropogenic contribution if climatology is used for biogenic aerosols, as indicated by Collin et al. (2011). Even though year-to-year trends in biogenic SOA follow year-to-year trends in fossil fuel organic carbon due to the characteristic of organic gas-aerosol partitioning, this interaction would not be accounted for if HadGEM uses climatology for biogenic aerosol. Again, clarification is needed.**

The inclusion of 'biogenicx5' in the legend of Figure 1b is an error and we have removed it. The previous version of the manuscript listed five species in the legend, but only showed four lines in the plot for the anthropogenic aerosol. We apologise for our carelessness, and are grateful that you picked this up.

**Figure 1b legend: All the lines in the legend are black; they should be colored to match the plotted lines.**

This legend has now been corrected, thank you.

**Minor Edits:**

**Page 5, line 6: 'increase in Asian emissions' -> 'increase in Asian anthropogenic emissions'.**

In line with our response to the first comment, we have changed this to 'Asian anthropogenic aerosol'.

**Page 6, line 3: 'Asian anthropogenic aerosol emissions' ->'increases in Asian anthropogenic emissions'.**

In line with our response to the first comment, we have changed this to 'Asian anthropogenic aerosol'.

**Page 6, line 13: 'can' -> 'could'**

Done.

[revised manuscript text omitted]

---

## Author Response (AR3)

Dear Nikos,

We are glad that you have found our responses to the reviewers to be thorough and satisfactory. We have now uploaded a revised manuscript that also addresses your own comments. We repeat your comments below in **bold**, along with our responses in plain text.

**1. How do other than high-level clouds, i.e. low- and mid-level ones, respond to the increase in Asian aerosol optical depth? Why do you emphasize high and not other clouds?**

We emphasise high clouds because they account for most of the cloud changes immediately downstream of Asia, and best reflect the circulation changes that occur there. As we already show in Figure 3 of the manuscript, the changes in total cloud amount (Figure 3b) in this region are dominated by the change in high clouds (Figure 3d). In the midlatitudes, there are some regions (western Europe most notably) where changes in high and low clouds are opposite in sign. We show changes in low, mid, and high level clouds alongside total cloud fraction below for completeness (Response Figure 1).

**2. What are the findings about statistical significance of trends at other than 10% levels, e.g. 5%? Has this been tested?**

We performed all t-tests at the 5% and 10% level. The choice of whether to use 5% or 10% to indicate significant results on a map is rather arbitrary, and the choice of the 5% level would have made little qualitative difference, even for a noisy variable like precipitation. Our indications of significance serve only to indicate to the reader where the result stands out from the noise. We consider the measure of agreement across our ensemble shown in Figure 6 to be a better, more physical, indicator of a forced response than simply showing significance at the 5% level instead of 10%.

We have included our temperature and precipitation results below (Response Figure 2) with three different masks. The first indicates where $p>0.1$ based on a two-sided t-test considering the ensemble mean (as shown in Figure 4 of the manuscript), the second where $p>0.05$, and the third where less than three ensemble members have a trend with the same sign as the ensemble mean trend (as shown in Figure 6 of the manuscript). In all three cases the same features are highlighted as being potentially interesting.

**3. Page 6, lines 10-11: reference is made to the longwave radiation, but associated results are not shown in the manuscript. Does the text really refer to the longwave radiation? If so, please make at least a reference to this, noting that relevant results are not shown in this paper.**

Done

**4. Please, add coordinates in maps (e.g. as done in Fig. 8a).**

Done

**5. In the second sentence of Figure 8 caption, indicate that text is referring to Fig. 8a. What about grey shading/solid contours in Fig. 8b? Please, clarify.**

Done

**6. Page 10, line 8: replace "? were found the have a global impact" by "? were found to have a global impact"**

Done

[Figure]

**Figure 1.** Ensemble mean (1998-2012)-(1979-1993) difference in (a): total cloud fraction, (b): high cloud fraction, (c): mid cloud fraction, and (d): low cloud fraction. Dots indicate where the response is not significant at the 10% level, as calculated using a two-tailed Student's t-test.

[Figure]

**Figure 2.** Ensemble mean (1998-2012)-(1979-1993) difference in (a): near-surface temperature and (b): precipitation. Dots indicate where the response is not significant at the 10% level, as calculated using a two-tailed Student's t-test. (c): near-surface temperature and (d): precipitation. Dots indicate where the response is not significant at the 5% level, as calculated using a two-tailed Student's t-test. (e): near-surface temperature and (f): precipitation. Dots indicate where the anomaly has the same sign in fewer than three ensemble members.

[revised manuscript text omitted]